# Erythrocyte Glutathione S-Transferase Activity as a Sensitive Marker of Kidney Function Impairment in Children with IgA Vasculitis

**DOI:** 10.3390/ijms25073795

**Published:** 2024-03-28

**Authors:** Marijan Frkovic, Ana Turcic, Alenka Gagro, Sasa Srsen, Sanda Huljev Frkovic, Dunja Rogic, Marija Jelusic

**Affiliations:** 1Department of Pediatrics, University Hospital Centre Zagreb, University of Zagreb School of Medicine, 10000 Zagreb, Croatia; 2Department of Laboratory Diagnostics, University Hospital Centre Zagreb, University of Zagreb Faculty of Pharmacy and Biochemistry, 10000 Zagreb, Croatia; 3Children’s Hospital Zagreb, Medical Faculty Osijek, Josip Juraj Strossmayer University of Osijek, 10000 Zagreb, Croatia; 4Department of Pediatrics, University Hospital Centre Split, School of Medicine, University of Split, 21000 Split, Croatia

**Keywords:** IgA vasculitis, nephritis, erythrocyte glutathione S-transferase, children

## Abstract

IgA vasculitis (IgAV) is the most common childhood vasculitis. The main cause of morbidity and mortality in children with IgAV is nephritis (IgAVN), but the risk of its development, severity, and chronicity remain unclear. Erythrocyte glutathione S-transferase (e-GST) activity has been previously detected as a sensitive marker of kidney function impairment in several diseases. We spectrophotometrically assessed and correlated e-GST activity between 55 IgAV patients without nephritis (IgAVwN), 42 IgAVN patients, and 52 healthy controls. At disease onset, e-GST activity was significantly higher in IgAVN patients (median (interquartile range)) (5.7 U/gHb (4.4–7.5)) than in IgAVwN patients (3.1 U/gHb (2.2–4.2); *p* < 0.001), and controls (3.1 U/gHb (1.9–4.2); *p* < 0.001). Therewithal, there were no differences between the IgAVwN patients and controls (*p* = 0.837). e-GST activity was also significantly higher in the IgAVN patients than in the IgAVwN patients after 3 months (5.0 U/gHb (4.2–6.2) vs. 3.3 U/gHb (2.3–4.1); *p* < 0.001) and 6 months (4.2 U/gHb (3.2–5.8) vs. 3.3 U/gHb (2.1–4.1); *p* < 0.001) since the disease onset. Consistent correlations between e-GST activity and serum creatinine, estimated glomerular filtration rate (eGFR), and proteinuria levels were not detected. In conclusion, increased e-GST activity can serve as a subtle indicator of kidney function impairment in children with IgAV.

## 1. Introduction

IgA vasculitis (IgAV) is the most common childhood vasculitis, with an incidence between three and 26.7 per 100,000 children [1,2,3,4]. According to the EULAR/PRINTO/PRESS Ankara 2008 criteria, IgAV is diagnosed in the presence of a non-thrombocytopenic palpable purpura or petechiae with lower limb predominance (mandatory), plus at least one of the following four criteria: abdominal pain, histopathological finding of leukocytoclastic vasculitis or proliferative nephritis with IgA deposits, arthritis or arthralgia, and proteinuria and/or hematuria as signs of kidney involvement [5]. IgAV is generally a benign, self-limiting disease [2,6,7]. The most common and serious complication, mostly due to a long-term prognosis, is nephritis (IgAVN), with an incidence of 20–60% [3,4,8,9,10] and extremely variable severity [11,12,13]. It is detected in up to 97% of all IgAVN patients within the 6-month period from IgAV onset [9,11,14], most commonly after 4–6 weeks of subcutaneous hemorrhage [15]. Most children with IgAVN have a good chance of recovery [15], whereas 1–15% of unpredictable cases develop chronic kidney disease (CKD) [9,10,16]. The factors that determine and mediate IgAVN are still not fully understood. To date, many attempts have been made to detect suitable genetic, histological, biochemical, or immunological markers of IgAVN development risk and its possible progression to CKD [10,16,17,18,19,20,21,22,23,24,25,26,27], but none have become a part of the routine diagnostic or follow-up evaluation in patients with IgAV [11].

Glutathione S-transferases are major phase II detoxification enzymes [28] and play an important role in the biotransformation of different xenobiotics and endobiotics, including active participation in the detoxification of uraemic toxins [29]. Their primary function is to catalyze the conjugation of electrophilic substrates to glutathione (GSH), with the purpose of their inactivation and dominant kidney excretion [29,30]. Human cytosolic GSTs are divided into seven subclasses with a tissue-specific arrangement [31]. The main GST in erythrocytes (e-GST) is a Pi class isoenzyme (GST-P), accounting for 90–95% of total erythrocyte GST activity [30,31]. Because the degree of e-GST expression is defined during erythrocyte maturation, and because the enzymatic content of mature erythrocytes remains unchanged throughout their whole life expectancy [31], it is assumed that, in patients with kidney function impairment, e-GST activity determined at a certain time represents the average exposure of erythroid cells to uremic toxins in the period of several weeks up to that time [31,32]. A significant increase in e-GST activity was detected in patients with kidney function impairment due to various causes. It has been described in patients with chronic kidney disease (CKD), with a positive correlation between enzyme activity and the stage of kidney disease [31,33]. In patients under maintenance hemodialysis, e-GST activity has been proposed as a novel marker of dialysis efficacy [31], as well as a marker of new organ excretory function in cases of kidney transplantation [32]. e-GST activity has also been proposed as a sensitive biomarker of early kidney damage in patients with diabetes mellitus type II (T2DM) [30] and systemic sclerosis (sSC) [34], and is more reliable than standardly used microalbuminuria and other common markers of kidney function [29].

In this study, we analyzed e-GST activity among children diagnosed with IgAV during the 6-month period since disease onset. We aimed to identify the differences in e-GST activity between patients with IgAV without nephritis (IgAVwN) and those with IgAVN, subsequently using it as a sensitive marker of kidney function impairment, i.e., IgAVN development risk and its possible progression to CKD.

## 2. Results

The study included 149 children under 18 years of age, 97 with a diagnosis of IgAV and 52 controls. The demographic and clinical findings of the children with IgAV and controls are presented in Table 1. The median age of 97 children with IgAV was 82 months, and the male-to-female ratio was 1.8:1. At disease onset, all 97 (100%) patients had skin lesions, 71 (73%) had arthralgia and/or arthritis, and 38 (39%) had abdominal pain. IgAVN was detected among 42 (43%) IgAV patients with a male-to-female ratio of 1.6:1.

Infection preceded the development of IgAV in 64 (65.9%) patients. A respiratory tract infection was reported in 54 (55.7%) patients and gastrointestinal tract infection in 10 patients (10.3%). In cases of a previous infection, the development of purpuric skin changes in patients with IgAVwN and IgAVN occurred for a median of 14 and 17 days, respectively.

IgAVN was detected in 34 (35%) patients with IgAV at disease onset, with isolated hematuria (>5 E/mm^3^) as the most common presentation (18/34, 53%), followed by a combination of hematuria and proteinuria (10/34, 29%) and only proteinuria (≥0.15 g protein/dU) (6/34, 18%). In 6 (6%) IgAV patients, IgAVN was discovered after three months with isolated hematuria as the most common presentation (5/6, 83%), followed by a combination of hematuria and proteinuria (1/6, 17%). In 2 (2%) IgAV patients, IgAVN was discovered 6 months after disease onset with isolated hematuria as the only presentation.

Among all IgAVN patients, 34 (81%) had laboratory signs of nephritis at disease onset, 25 (59%) after 3 months, and 18 (43%) after six months since disease onset. Two (5%) of all IgAVN patients diagnosed with IgAVN had temporarily normalized urine findings (abnormal findings at disease onset and 6 months after disease onset and normal findings after 3 months after disease onset). Twelve (29%) patients with IgAVN had laboratory signs of nephritis during the study period.

Except for the significantly longer time elapsed from the previous infection to the development of skin changes in patients with IgAVN, there were no significant differences between IgAVwN and IgAVN patients regarding age, gender, or frequency of initial symptoms (Table 1).

Among routine laboratory findings, both IgAVN and IgAVwN patients had significantly higher ESR, CRP, leukocytes, and platelet concentrations than the control group. By comparing the IgAVN and IgAVwN patients at disease onset, significantly higher serum creatinine and IgA concentrations, E/mm^3^ of urine, and urinary protein excretion, in addition to lower eGFR, were observed in IgAVN patients than in IgAVwN patients. (Table 1).

While comparing IgAVN and IgAVwN patients during the six-month follow-up period, significantly higher values of serum creatinine concentration and E/mm^3^ of urine, as well as significantly lower values of calculated eGFR, were consistently detected in IgAVN patients, but serum creatinine concentration and calculated eGFR were within the referral boundaries in both groups of patients during the whole follow-up period. Also, a significant decrease/normalization in ESR, CRP, leukocytes, and serum IgA and IgM concentrations were detected using the dependent value changes analysis in all IgAV patients during the six-month follow-up period (*p* < 0.001/*p* < 0.001 for all values) (Appendix A).

At the IgAV onset, e-GST activity was significantly higher in IgAVN patients: median (interquartile range) 5.7 U/gHb (4.4–7.5 U/gHb) compared with the activity in IgAVwN patients 3.1 U/gHb (2.2–4.2 U/gHb); *p* < 0.001, and controls 3.1 U/gHb (1.9–4.2 U/gHb); *p* < 0.001. Simultaneously, there were no differences in e-GST activity between patients with IgAVwN and controls (*p* = 0.837) (Figure 1).

e-GST activity was also significantly higher in IgAVN patients compared with IgAVwN patients after three (5.0 U/gHb (4.2–6.2) vs. 3.3 U/gHb (2.3–4.1); *p* < 0.001) and six months (4.2 U/gHb (3.2–5.8) vs. 3.3 U/gHb (2.1–4.1); *p* < 0.001) since the disease onset (Figure 2).

No significant differences in e-GST activity were observed between IgAVN patients with initially normal urinary findings and IgAVN patients with initially abnormal urinary findings (Appendix A).

Consistent correlations between e-GST activity and demographic, clinical, and laboratory variables, including serum creatinine, eGFR, or proteinuria, were not detected in our study. A positive correlation was detected only between e-GTS activity and E/mm^3^ of urine in patients with IgAVN 3 and 6 months after disease onset (Appendix A).

In the receiver operating characteristic (ROC) analysis of the e-GST activity in the prediction of nephritis occurrence, at the beginning of the study, a significant area under the curve (AUC) of 91.2% (95% CI: 85.8–96.5%, *p* < 0.001 against null model without predictors) was found with a sensitivity of 90.5% and specificity of 72.7% at the value of e-GST > 4.15 U/gHb. Furthermore, ROC analysis of the nephritis predictivity with the hematuria model resulted with an AUC of 87.2% (95% CI: 79.4–95.1%, *p* < 0.001), sensitivity of 76.2%, and specificity of 96.4% at the value of >3.5 erythrocytes in urine. The AUC of the e-GST model, compared to the AUC of the hematuria model, was not significantly different (95% CI of AUC difference: (−5.8)–13.7%, *p* = 0.432). ROC analysis of the proteinuria model resulted with the AUC of 69.0% (95% CI: 58.2–79.8%, *p* = 0.001), sensitivity of 40.5%, and specificity of 98.2% at the value of >0.14 g/L of urine. The AUC of the e-GST model, compared with the AUC of the proteinuria model, was significantly greater (95% CI of AUC difference: 10.9–33.4%, *p* < 0.001) (Figure 3).

Similarly, in the ROC analysis of the e-GST activity in the prediction of nephritis occurrence, after 3 months since the disease onset, a significant area under the curve (AUC) of 85.4% (95% CI: 77.4–93.3%, *p* < 0.001 against null model without predictors) was found with a sensitivity of 73.8% and specificity of 83.6% at the value of e-GST > 4.35 U/gHb. Furthermore, ROC analysis of the nephritis predictivity with hematuria model resulted with the AUC of 83.7% (95% CI: 76.0–91.3%, *p* < 0.001), sensitivity of 71.4%, and specificity of 90.9% at the value of >0.54 erythrocytes in urine. The AUC of the e-GST model, compared to the AUC of the hematuria model, was not significantly different (95% CI of AUC difference: (−6.8)–10.2%, *p* = 0.694). ROC analysis of the proteinuria model resulted with the AUC of 61.5% (95% CI: 50.1–72.8%, *p* = 0.048), sensitivity of 57.1%, and specificity of 61.8% at the value of >0.08 g/L of urine. The AUC of the e-GST model, compared with the AUC of the proteinuria model, was significantly greater (95% CI of AUC difference: 10.8–37.0%, *p* < 0.001) (Figure 4).

Analogously, in the ROC analysis of the e-GST activity in the prediction of nephritis occurrence, after 6 months since the disease onset, a significant area under the curve (AUC) of 72.0% (95% CI: 61.3–82.6%, *p* < 0.001 against null model without predictors) was found with a sensitivity of 45.2% and specificity of 94.5% at the value of e-GST > 4.75 U/gHb. Furthermore, ROC analysis of the nephritis predictivity with hematuria model resulted with the AUC of 77.3% (95% CI: 68.3–86.3%, *p* < 0.001), sensitivity of 61.9%, and specificity of 85.5% at the value of >1.51 erythrocytes in urine. The AUC of the e-GST model, compared to the AUC of the hematuria model, was not significantly different (95% CI of AUC difference: (−15.7)–5.1%, *p* = 0.317). ROC analysis of the proteinuria model resulted with the AUC of 60.5% (95% CI: 49.2–71.5%, *p* = 0.050), sensitivity of 97.6%, and specificity of 21.8% at the value of >0.05 g/L of urine. The AUC of the e-GST model, compared with the AUC of the proteinuria model, was not significantly different (95% CI of AUC difference: (−1.7)–24.9%, *p* = 0.088) (Figure 5).

## 3. Discussion

Nephritis is the most common, and almost the only, cause of morbidity and mortality in patients with IgAV [2,6,9,11,13,14,35].

Routinely used markers of kidney function (BUN and serum creatinine, eGFR) in children with IgAV are generally within the referral boundaries, except in extremely rare cases of severe, acute nephritis with nephrotic or nephritic syndrome at disease onset or cases of prolonged, untreated chronic nephritis. Standard urinary tests seem to be insufficiently specific and sensitive for the assessment of IgAVN development and activity because, according to some authors, even normal urinary findings do not exclude the possibility of nephritis activity and chronic IgAVN development [9,36]. Kidney biopsy is an invasive procedure with possible complications and histological findings that indicate current kidney changes without nephritis occurrence or progression risk [37].

The non-existence of specific and sensitive kidney function impairment markers can lead to serious misjudgments in terms of possible over- or under-treatment of patients with IgAV [38].

IgAVN was detected in 43.3% of our IgAV patients during the 6-month follow-up period, which fits within the wide range of nephritis occurrence listed between 20% and 60% of all IgAV patients under 18 years of age [4,8,9,10]. Among IgAVN patients, 81% had laboratory signs of nephritis at disease onset, 59.5% after three months, and 42.9% after six months since disease onset.

The proportion of nephritis initially detected among all patients with IgAVN in our study was closest to the results of Zhang et al., who detected nephritis in 85% of all patients with IgAVN within the four-week period after the appearance of the first skin lesions [39]. However, in some other studies, nephritis was usually detected 4–6 weeks or even a few months after IgAV onset [11,40,41]. Differences in the obtained results are probably due to the inconsistency of the applied criteria for proteinuria and hematuria (commonly milder criteria compared to our criteria in most other studies) [10,38]. As opposed to these results, a decrease in the number of IgAVN cases during the 6-month follow-up and isolated hematuria as the only sign of nephritis in most IgAVN patients in our study are consistent with the usually benign and mild course of nephritis described in most studies [11,40,42,43].

Despite the abovementioned findings, several authors point out the possibility of transient improvement or normalization of urine findings in some IgAVN patients and suggest a longer follow-up period than six to twelve months proposed in the standard Single Hub and Access point for pediatric Rheumatology in Europe) guidelines for IgAV [2,9,11,36,44].

Compared with the control group, both groups of patients with IgAV had significantly higher ESR, CRP, and leukocyte concentrations at disease onset. While comparing other laboratory findings among patients with IgAVN and IgAVwN at disease onset, significant differences in our research were detected only in serum IgA concentrations (significantly higher concentrations in patients with IgAVN; *p* 0.008/<0.001). During the follow-up period, a significant decrease/normalization of ESR, CRP, leukocyte, and serum IgA concentrations were detected by the dependent value changes analysis in all patients with IgAV. Higher ESR, and CRP, leukocyte, and IgA concentrations in all patients with IgAV indicate a systemic inflammatory event/immune reaction as the basis for the development of IgAV symptoms. Their normalization in all patients during the follow-up period correlates with the regression of extrarenal symptoms, indicating a regression of systemic inflammation. On the one hand, these results are in accordance with the previously known, but pathogenetically still unclear, self-limiting nature of IgAV. On the other hand, their normalization in IgAVN patients indicates that nephritis, as a complication of the underlying disease, probably arises because of protracted/continued, but dominantly local, immunological events without repercussions for the above findings.

During the follow-up period, significantly higher serum creatinine concentration, as well as significantly lower values of calculated eGFR, were consistently detected in IgAVN patients compared with IgAVwN patients. In our opinion, despite the significant differences obtained in our research, serum creatinine, calculated eGFR, and IgA concentrations are inappropriate for distinguishing between patients with IgAVN and IgAVwN because all values of these findings were within the referral boundaries in both groups at all three points of determination, and thus did not reach clinical importance.

In our study, infection preceded the development of the first symptoms of IgAV in more than 60% of patients. The registered time elapsed from the previous infection to the appearance of purpura, which was significantly longer in patients with IgAVN, was the only parameter that distinguished patients with IgAVwN from those with IgAVN regarding the previous infection in the anamnesis. Although these data indicate the importance of environmental factors in the presumed pathogenesis of IgAV, they do not contribute to the prediction of nephritis development.

All these results highlight the need to identify appropriate new markers for the detection of patients with IgAV at risk of developing nephritis [45,46].

In earlier studies by other authors, the activity of e-GST was primarily studied in adult patients with a spectrum of kidney diseases. Because e-GST activity was described as a sensitive marker of kidney disease severity in CKD patients, with potential use in the assessment of dialysis efficacy and new organ excretory function in cases of kidney transplantation (including cases of acute transplant rejection), as well as a possible marker of early kidney damage in cases of T2DM and sSC [30,31,32,33,34], we assumed that it would also serve as a sensitive marker of kidney function impairment in children with IgAV, as well as a non-immunological marker of kidney immunological process activity in children with IgAVN.

In all patients included in our study, e-GST activity was determined at the time of study inclusion and in all IgAV patients at 3 and 6 months after inclusion. The median activity of e-GST among controls was 3.13 U/gHb, which is between the results obtained by Anastasov et al. (2.0 ± 0.53 U/gHb) [47] and Orhan et al. (4.3 ± 0.2 U/gHb) [48], although some authors reported higher average activities [33,49]. Differences between normal average e-GST activity obtained in various studies are primarily attributed to polymorphisms in e-GST genes in certain populations of healthy people [50]. The initial e-GST activity in our IgAVwN patients was not significantly different from that in controls, nor did it change significantly during the 6-month follow-up period. In contrast, the initially determined e-GST activity in our IgAVN patients was significantly higher than that in IgAVwN patients and controls and remained significantly higher in IgAVN patients than in IgAVwN patients during the 6-month follow-up period. Our results not only suggest the preservation of kidney excretory functions in our IgAVwN patients but, combined with the positive correlation registered only between e-GST activity and E/mm^3^ of urine after 3 and 6 months since the disease onset in our IgAVN patients, support the high sensitivity of e-GST activity in detection of even the slightest kidney function decline described in several previous studies [29,30,32,33,34], since isolated hematuria in IgAVN is considered a much milder repercussion of kidney autoimmune process compared with proteinuria [42]. The results obtained in our study are consistent with the studies on patients with CKD carried out by Dessì et al. and Noce et al., in which no correlations were found between e-GST activity and markers of systemic inflammation or kidney function. Contrary to our assumption that nephritis represents only a protracted, local immune event, these authors assume that not all uraemic toxins have the same systemic inflammation trigger potential and that e-GST can inactivate many uraemic toxins, regardless of their inflammatory potential [31,33]. No correlation was found between e-GST activity and serum creatinine, eGFR, and proteinuria in our study. These results are consistent with the results of Fabrini et al. who reported increased e-GTS activity in sSC patients at early stages of kidney damage without repercussions on eGFR [34], as well as with the results of Bocedi et al., Noce et al., and Tesauro et al., who registered increased e-GST activity in T2DM patients without the usual laboratory signs of nephropathy development [29,30].

In addition to the first description of elevated e-GST activity registered during follow-up in children with IgAVN, we also describe a significantly elevated enzyme activity in the early, acute phase of the disease. Similar acute changes in e-GST activity have been previously described only by Bocedi et al. in cases of acute rejection of kidney transplants in adult patients [32].

Because e-GST activity determined at a particular time point reflects the average impact of enzyme induction molecules in erythroid precursors during the few previous weeks [31,32], significantly higher e-GST activity at the disease onset in our IgAVN patients, given the relatively short time (median: 17 days) from the previous infection as a presumed trigger of the immunological process to the first appearance of skin efflorescence, possibly indicates intense local (kidney) immunological changes at the very early stages of systemic immune response, which quickly lead to otherwise routinely undetected kidney function impairment.

However, the time span between the previous infection and purpura onset in our IgAVN patients seemed to be long enough for significant intense changes in e-GST activity. An alternative explanation for the increased e-GST activity in children with IgAVN would be exposure to an exogenous, so far unidentified toxin that, in addition to influencing e-GST activity, modulates and stimulates the presumed sequence of immune events in the kidneys of children with IgAV with the consequent development of nephritis, i.e., makes the kidneys of children with IgAVN more vulnerable to immunological events compared with children with IgAVwN. This opens the possibility of undiscovered differences in the etiopathogenesis of IgAVN and IgAVwN.

The limitations of this study are the relatively short follow-up period and relatively small number of patients, especially in the subgroup with later nephritis onset. The extension of the future study period and increase in patient number are necessary to reveal the full potential of e-GST activity in the assessment of patients with IgAV.

## 4. Materials and Methods

### 4.1. Design

This prospective study was conducted at the Department of Pediatrics, University of Zagreb School of Medicine, University Hospital Centre Zagreb, Croatia, with the approval of the Ethics Committee of the University Hospital Centre Zagreb. Parental written consent was obtained from all patients. IgAV was diagnosed on the basis of the ELAR/PRINTO/PRESS Ankara 2008 criteria [5]. Patients with a history of previous kidney or systemic diseases were excluded. Among all IgAV patients, the presence of haematuria (>5 erythrocytes per mm^3^/urine) and/or proteinuria (≥0.15 g/dU) at disease onset and/or 3 and/or 6 months after disease onset defined the IgAVN subgroup. Controls were recruited from children without any clinical or laboratory signs of inflammation or kidney disease.

### 4.2. Methods

Clinical data, including gender and age at disease onset and recent infections before the appearance of purpura, were obtained for each subject.

Blood samples were obtained for erythrocyte sedimentation rate (ESR), C-reactive protein (CRP), complete blood cell count, blood urea nitrogen (BUN), serum creatinine, total serum proteins, serum immunoglobulin (Ig) A, IgM, IgG, and C3 and C4 complement component concentrations. First, urine samples were obtained for erythrocyte count per mm^3^ (E/mm^3^ of urine) and 24 h urine samples for the quantification of urinary protein excretion. Estimated glomerular filtration rates (eGFR) were calculated using the bedside Schwartz equation. Stool samples were obtained for microscopic blood traces. Blood, urine, and stool samples were collected from all subjects at disease onset/study inclusion. Additional blood, urine, and stool samples were collected only from patients with IgAV at 3 and 6 months after disease onset.

For the determination of e-GST activity, 3 mL of blood in EDTA was stored at 4 °C for no longer than 72 h. The whole blood sample was washed, and the buffy coat was removed three times with saline by centrifugation at 1000× *g* for 5 min. Hemoglobin concentrations were measured using a Sysmex XE 5000 analyzer (Sysmex Europe SE, Norderstedt, Germany). Hemolysate was obtained by diluting one volume (40 μL) of blood in 25 volumes (1 mL) of cold bi-distilled water. e-GST activity was assessed spectrophotometrically at 340 nm according to the manufacturer’s instructions (Glutathione S-transferase (GST) Assay Kit, CS0410, Sigma Aldrich, St. Louis, MO, USA). Briefly, in a quartz cuvette containing 1 mL of substrate solution (980 μL Dulbecco’s phosphate-buffered saline, 10 μL of 200 mM reduced L-glutathione and 10 μL of 100 mM 1-chloro-2,4-dinitrobenzene (CDNB)) at 37 °C, used as a blank, a 50 μL of hemolyzed sample was added. Absorbance was read every 30 s for 5 min after a lag time of 1 min. e-GST activity was expressed as enzyme units (U) per gramme of Hb. One unit represents the e-GST activity that catalyzes the formation of 1 μmole CDNB-glutathione conjugate in 1 min at 37 °C.

Statistical analyses were performed using IBM SPSS Statistics, version 25.0 (https://www.ibm.com/analytics/spss-statistics-software, accessed on 11 January 2020.) and MedCalc for Windows, version19.0.3 (MedCalc Statistical Software version 19.0.3 (MedCalc Software bvba, Ostend, Belgium; https://www.medcalc.org; 2019)). Median (interquartile range, IQR) values and frequencies are provided for continuous and categorical variables. Continuous values were compared using the Kruskal–Wallis test followed by post hoc comparison using the Mann–Whitney U test. Fischer’s exact test was used to analyze differences in categorical variables. Spearman’s rank correlation coefficient was used to assess the correlation between the two continuous variables. All significant results in bivariate analysis were used in stepwise multivariate analysis (binary logistic regression) to assess multivariate prediction of nephritis development. For e-GST, we constructed an ROC curve and calculated the area under the curve (AUC) to evaluate the significant cut-off value with the highest predictive accuracy to discriminate kidney involvement. All values of *p* < 0.05 were considered statistically significant.

## 5. Conclusions

In this study, e-GST activity was significantly higher in children with IgAVN than in children with IgAVwN at disease onset and healthy children. No significant differences in e-GST activity were observed between IgAVN patients with initially normal urinary findings and IgAVN patients with initially abnormal urinary findings.

In children with IgAVN, e-GST activity gradually decreased 3 and 6 months after IgAV onset, but remained significantly higher than that in children with IgAVwN.

In children with IgAVwN, initially determined e-GST activity was not significantly different from that in healthy children, nor did it change significantly during follow-up.

The activity of e-GST 4.1 U/gHb determined at disease onset, with a sensitivity of 90.5% and a specificity of 72.7%, detected IgAV patients at risk of developing IgAVN. When determining the activity of e-GST 3 and 6 months after IgAV onset, the sensitivity decreased and the specificity increased, indicating that e-GST had the greatest value in predicting the development of IgAVN at the time of IgAV onset. According to our study, eGST activity is not inferior to erythrocyturia, and it is superior to proteinuria in the prediction of nephritis in patients with IgAV.

Serum e-GST activity is a sensitive and specific, minimally invasive, easily performed, and inexpensive laboratory test. It can therefore serve as an additional reliable biomarker for assessing the risk of IgAVN development, particularly in patients with initially normal urine findings, temporary normalization of urine findings or prolonged microhematuria and/or borderline proteinuria during follow-up.

## Figures and Tables

**Figure 1 ijms-25-03795-f001:**
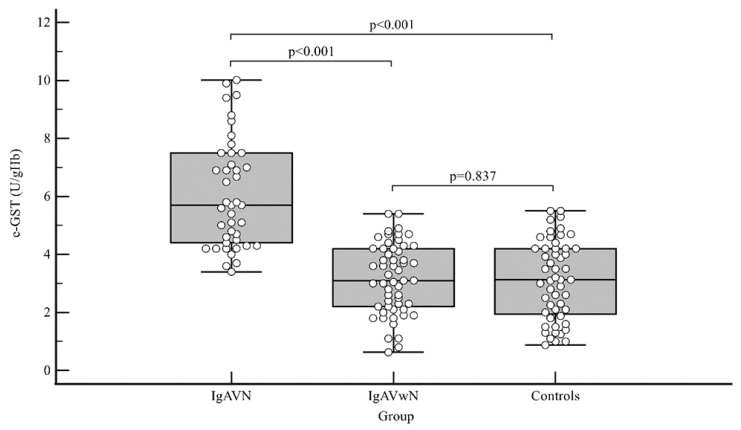
Erythrocyte glutathione S-transferase (e-GST) activity differences between IgA vasculitis without nephritis (IgAVwN), IgA vasculitis nephritis (IgAVN) patients, and control group at the beginning of the study: *p* < 0.001 (Kruskal–Wallis test).

**Figure 2 ijms-25-03795-f002:**
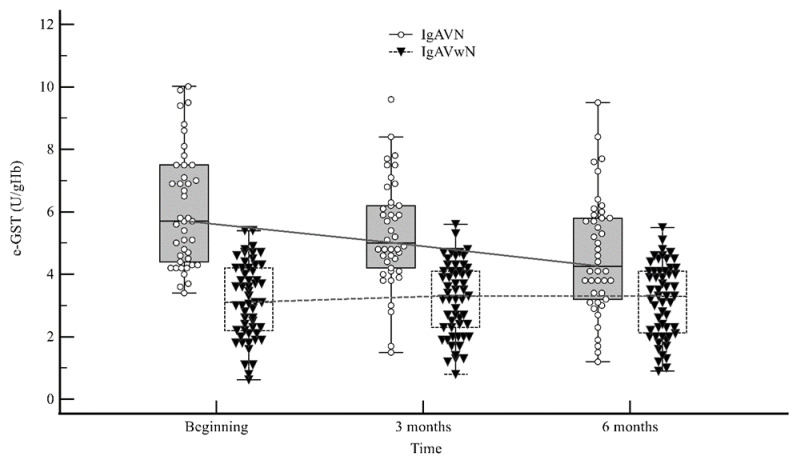
Dependent changes of erythrocyte glutathione S-transferase (e-GST) activity in IgA vasculitis without nephritis (IgAVwN) and IgA vasculitis nephritis (IgAVN) patients during the six-month follow-up period.

**Figure 3 ijms-25-03795-f003:**
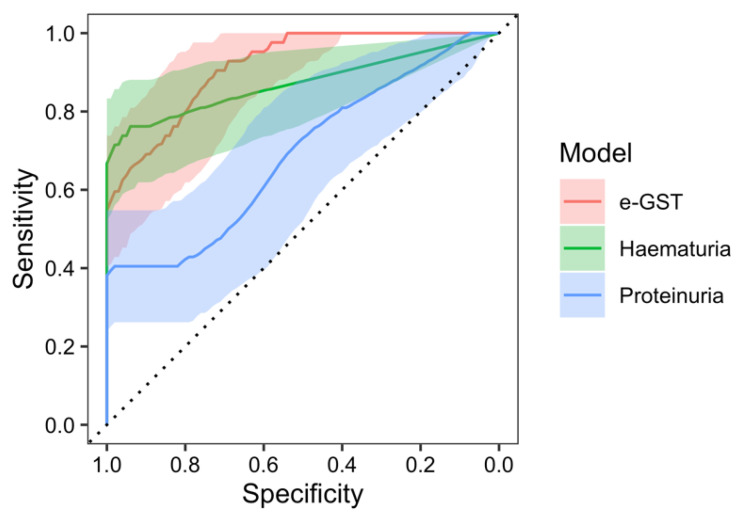
ROC analysis of erythrocyte glutathione S-transferase (e-GST) activity (red), hematuria (green), and proteinuria (blue) models in the prediction of nephritis development at the beginning of the study.

**Figure 4 ijms-25-03795-f004:**
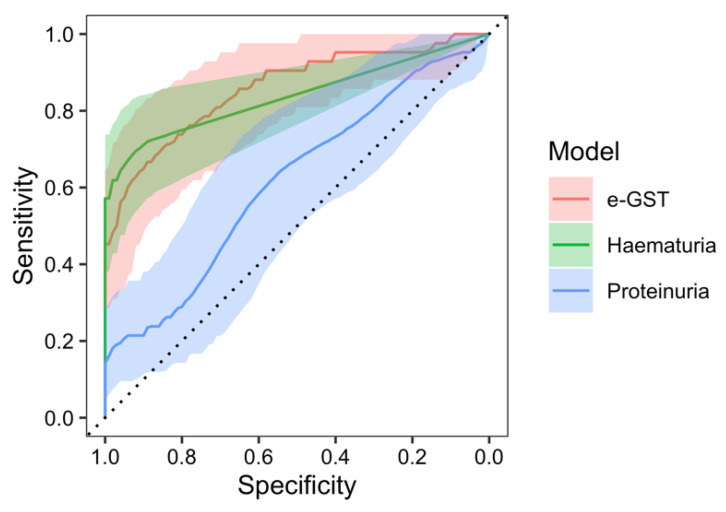
ROC analysis of erythrocyte glutathione S-transferase (e-GST) activity (red), hematuria (green), and proteinuria (blue) models for the prediction of nephritis 3 months after disease onset.

**Figure 5 ijms-25-03795-f005:**
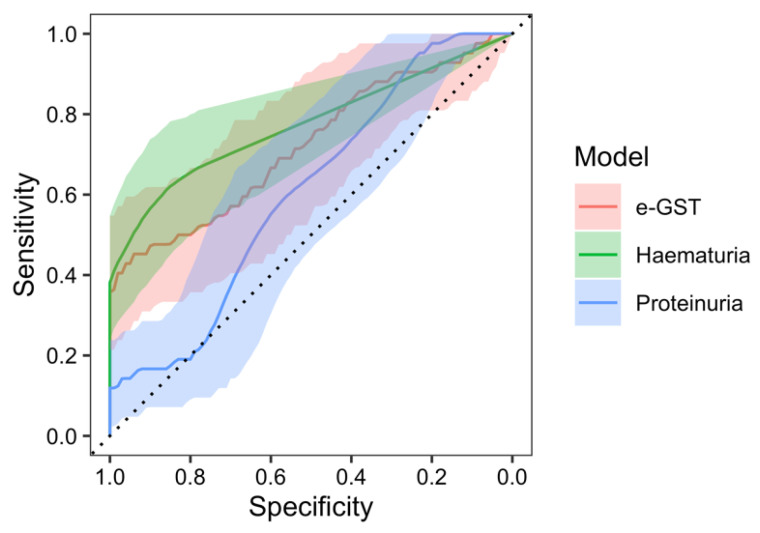
ROC analysis of erythrocyte glutathione S-transferase (e-GST) activity (red), hematuria (green), and proteinuria (blue) models for predicting nephritis 6 months after disease onset.

**Table 1 ijms-25-03795-t001:** Demographic, clinical and laboratory variables of patients with IgAVwN, IgAVN, and controls at the beginning of study.

	IgAVwN	IgAVN	Controls
Age (months): median (IQR)	72.0 (58.0–107.0)	91.0 (56.3–138.0)	115.0 (76.3–138.5)
Gender (M:F)	36:19	26:16	36:16
Previous infection, n (%)	38 (69.1)	26 (61.9)	-
Time between infection and appearance of purpura (days): median (IQR) *	14.00 (10.00–17.25)	17.00 (11.75–24.25)	-
Purpura: n (%)	55 (100.0)	42 (100.0)	-
Arthritis/arthralgia: n (%)	44 (80.0)	27 (64.3)	-
Abdominal pain: n (%)	22 (40.0)	16 (38.1)	-
Nephritis initially: n (%)	0 (0.0)	34 (81.0)	-
ESR (mm/h): median (IQR) **^,^***	18.00 (12.00–25.00)	17.00 (10.00–24.00)	7.00 (5.00–11.00)
CRP (g/L): median (IQR) **^,^***	8.00 (2.03–14.93)	6.05 (1.58–15.98)	0.30 (0.30–0.30)
Hb (g/L): median (IQR) **	127.00 (121.00–132.00)	130.50 (122.00–136.00)	132.00 (126.25–138.75)
L (×10^9^/L): median (IQR) **^,^***	11.30 (8.58–14.81)	11.67 (8.93–14.18)	7.40 (5.93–9.13)
Plt (×10^9^/L): median (IQR) **^,^***	342.00 (278.00–427.00)	388.50 (314.50–441.75)	294.00 (253.00–329.00)
BUN (mmol/L): median (IQR)	3.90 (3.20–4.70)	4.15(3.48–5.05)	4.10 (3.30–4.98)
Serum creatinine (μmol/L): median (IQR) *^,^**	38.00 (30.00–48.00)	45.00 (38.25–53.00)	44.00 (39.00–48.00)
Feritin (μg/L)	65.60 (36.13–102.10)	68.00 (46.50–93.20)	59.80 (42.78–98.75)
Total serum proteins (g/L): median (IQR) **	68.00 (64.00–71.00)	70.50 (65.00–74.00)	71.00 (69.00–73.00)
IgG (g/L): median (IQR)	10.03 (8.20–11.78)	10.10 (8.18–12.41)	10.32 (9.01–12.01)
IgM (g/L): median (IQR) *^,^**	0.79 (0.63–1.04)	0.94 (0.76–1.22)	0.98 (0.66–1.32)
IgA (g/L): median (IQR) *^,^***	1.65 (1.25–2.15)	2.11 (1.64–2.85)	1.58 (1.12–2.07)
E/mm^3^ of urine: median (IQR) *^,^***	0.00 (0.00–1.00)	9.50 (3.50–36.25)	0.00 (0.00–2.00)
24 h urinary protein excretion (g/dU): median (IQR) *^,^***	0.07 (0.05–0.10)	0.09 (0.07–0.21)	0.07 (0.04–0.11)
eGFR (mL/min/1.73 m^2^): median (IQR) *	121.06 (94.26–138.22)	99.81 (89.65–118.26)	111.40 (99.00–120.24)
Positive fecal occult blood test: n (%)	7 (12.7)	9 (21.4)	0.0

Legend: * IgAVwN vs. IgAVN, *p* < 0.05, ** IgAVwN vs. controls, *p* < 0.05, *** IgAVN vs. controls, *p* < 0.05, IgAVwN—IgA vasculitis without nephritis, IgAVN—IgA vasculitis nephritis, IQR—interquartile range, ESR—erythrocyte sedimentation rate, CRP—C-reactive protein, Hb—hemoglobin, L—leucocytes, Plt—platelets, IgG—immunoglobulin G, IgA—immunoglobulin A, IgM—immunoglobulin M, E—erythrocytes, eGFR—estimated glomerular filtration rate.

## Data Availability

All data are available in the manuscript. The datasets generated during and/or analyzed during the current study are available from the corresponding author on reasonable request.

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
