# Peer review of "Erythrocyte Glutathione S-Transferase Activity as a Sensitive Marker of Kidney Function Impairment in Children with IgA Vasculitis"

_ijms, 2024, doi:10.3390/ijms25073795_

Round 1
Reviewer 1 Report
Comments and Suggestions for Authors
This is a study of 97 patients with IgA vasculitis (IgAV) to evaluate the erythrocyte glutathione S-transferase (e-GST) activity. The authors demonstrated that e-GST activity levels were significantly elevated in the group of IgAV with nephritis (IgAVN) than those in groups of IgAV without nephritis (IgAwN) and controls.
The presented study was well performed, and the manuscript is described in a reasonable manner. However, the analysis results seem to be insufficient. I am not sure whether this e-GST activity is more suggestive of the presence of nephritis than other indicators (urinary protein excretion, or erythrocytes of urine), or whether it is an index reflecting the activity and prognosis of nephritis.
There is a description in the method section that the Area Under the Curve (AUC) and the cut off value of the e-GST activity for nephritis were calculated using the Receiver Operating Characteristic (ROC) curve, but I could not find those results in the result section. Those data of the e-GST activity are one of the important results of this study, so authors should provide the ROC curve with the cut-off value and AUC for nephritis by adding other parameters including estimated glomerular filtration rate (eGFR), urinary protein excretion, and erythrocytes of urine.
Moreover, there are other points that I would like to inquire as follows.
1. Why was the serum creatinine level significantly lower in the IgAwN group than in the control group?
2. In addition to e-GST activity, serum IgA levels were also higher in the IgAVN group compared to the other two groups. Although there was no correlation between urine protein excretion, or eGFR and e-GST activity, was there a correlation between the GST activity and serum IgA levels? Also, serum C4 levels were lower in the IgAVN group compared to the other two groups, but what about this? I think it would be better to provide that there is no relationship between e-GST activity and characteristics (gender, age, or etc.) or other clinical indicators (all indicators listed in the Table 1).
3. I think Table 2 should be simplified, presented in supplemental file, or deleted. On the other hand, there is no correlation between eGFR and e-GST, but please provide two-dimensional diagrams of both parameters in the supplementary file.
4. Please provide the ROC curve as described above.
Author Response
Dear Sir/Madam
Thank you for your efforts and suggestions regarding our study on e-GST activity in children with IgA vasculitis.
With this research, we aimed to prove that e-GST activity can be used as an additional, sensitive marker of nephritis development and activity, especially in children with initially normal urine findings (8 children in our research developed proteinuria and/or haematuria after 3- or 6-month period since the disease onset) and in children with transitory normalisation of urine findings during follow-up. We also implicate that children with IgAVN and continuously increased activity or progressive increase in e-GST activity are at risk of developing chronic nephritis, independent of urine findings. To provide additional clarifications, we have expanded the Discussion section and added an evaluative list of conclusions.
- Our control group was significantly older than the group of patients with IgAV and had a male predominance. Because e-GST activity remains stable during almost the entire life expectancy and is considered to be independent of gender, we considered that the given difference will not affect e-GST as the main parameter of our research. In contrast, creatinine reference values ​​increase with age, and during adolescence, they are higher in males than in females. We believe that this is the reason for the difference in serum creatinine concentration between patients with IgAVwN and the control group.
- No correlations were detected between serum IgA levels and e-GST activity. Higher levels of IgA concentrations in patients with IgAVN can be a sign of more intensive immunological event, but in our opinion, despite the significant differences obtained in our research, IgA concentrations (as well as serum creatinine, calculated eGFR) are inappropriate for distinguishing between patients with IgAVN and IgAVwN because all values of these findings were within the referral boundaries in both groups at all three points of determination and thus did not reach the clinical importance. The same applies to the initial values ​​of C4. However, as we did not determine the values ​​of the complement components in the further course of the research, we have omitted the complement findings in the revised manuscript. We have added the abovementioned comprehensive explanation of our opinion to the Discussion section and 2 tables with routine laboratory findings in patients 3 and 6 months after disease onset.
- Table 2. is simplified and is planned to be replaced in the supplementary file.
- ROC analysis of erythrocyte glutathione S-transferase (e-GST) activity in the prediction of nephritis development has been added to the main text as suggested.
Reviewer 2 Report
Comments and Suggestions for Authors
I considered the manuscript entitled “Erythrocyte glutathione S-transferase activity as a sensitive marker of kidney function impairment in children with IgA vasculitis” by Marijan Frkovic, et al that is intended to be published in IJMS journal.
I really enjoyed the manuscript narrative. it is well planned and exquisitely presented, in a population of children affected of a rare disease. However, to me the value of eGST levels in the disease should appear as unnecessary for every day clinics.
There are no differences between proteinuria in IgAVwN and IgAVN. This is a somewhat surprise since both groups should appear highly different in terms of renal affectation. How do you explain this? There are 81% of the patients with nephritis at onset, had these patients proteinuria? Proteinuria is a frequent manifestation of clear nephritis. Or you just define nephritis in these patients who only has isolated hematuria? You really have not patients with moderate or severe nephritis. This may be also reflected by the low increase of creatinine, from 38 to 45 umol/L representing slight renal failure. A Table or Figure with the follow-up values for creatinine, GFR, urine erythrocytes and proteinuria should be added. What about eGST in patients with severe nephritis?
What about kidney biopsies in patients with nephritis? You speak about morbidity and mortality in IgAVN but this does not seem the case of your population, as your patients appear more or less healthy.
In fact, the measurement of eGST appear a marginal biomarker in your population. With the scarce renal manifestations, what is the significance of eGST? What is the advantage of measuring eGST against renal biopsy or GFR? in slight or in severe cases of nephritis? What is the benefit of eGST compared to isolated hematuria? This is, if a patient has hematuria, then it has nephritis… If a patient has elevated eGST, then he has nephritis….? Physicians will continue measuring hematuria?
The conclusion: e-GST activity can serve as a sensitive marker of kidney function impairment at the disease onset and initial follow-up period in children with IgAV, appears weak. Again, which is the advantage of measuring eGST against creatinine?
Author Response
Dear Sir/Madam
Thank you for your efforts and suggestions regarding our study on e-GST activity in children with IgA vasculitis.
With this research, we aimed to prove that e-GST activity can be used as an additional, sensitive marker of nephritis development and activity, especially in children with initially normal urine findings (8 children in our research developed proteinuria and/or haematuria after 3- or 6-month period since the disease onset) and in children with transitory normalisation of urine findings during follow-up. We also implicate that children with IgAVN and continuously increased activity or progressive increase in e-GST activity are at risk of developing chronic nephritis, independent of urine findings. To provide additional clarifications, we have expanded the Discussion section and added an evaluative list of conclusions.
We have defined IgAVN with urine findings of >5 E/mm³, and/or proteinuria of ≥0,15 g protein/dU. IgAVN was detected in 34 (35%) patients with IgAV at disease onset, with isolated haematuria (>5 E/mm³) as the most common presentation (18/34, 53%), followed by a combination of haematuria and proteinuria (10/34, 29%) and only proteinuria (≥0,15 g protein/dU) (6/34, 18%). In 6 (6%) IgAV patients, IgAVN was discovered after three months with isolated haematuria as the most common presentation (5/6, 83%), followed by a combination of haematuria and proteinuria (1/6, 17%). In 2 (2%) IgAV patients, IgAVN was discovered 6 months after disease onset with isolated haematuria as the only presentation. Among all IgAVN patients, 34 (81%) had laboratory signs of nephritis at disease on-set, 25 (59%) after 3 months, and 18 (43%) after six months since disease onset. Twelve (29%) patients with IgAVN had laboratory signs of nephritis during the study period.
During the 3-year period of our study, we did not have a single patient with severe acute or subacute nephritis, massive haematuria, proteinuria over 1 g/dU, significant increase in serum creatinine or BUN, or eGFR decline. We performed kidney biopsy only in three children with IgAV due to proteinuria between 0.5 and 1 g/dU. In the patients who underwent biopsy, the pathohistological findings corresponded to nephritis as part of IgAV. This is not unusual for a population of patients with IgAV. Therefore, in cases of severe nephritis, we can only speculate about the dynamics and levels of e-GST activity. In these cases, regular findings would probably clearly indicate nephritis. To provide additional clarifications of our findings, we have added 2 tables with routine laboratory findings in patients 3 and 6 months after disease onset.
In introduction and discussion, we have pointed out that most children with IgAVN have a good chance of recovery, whereas 1-15% of unpredictable cases develop chronic kidney disease (CKD). Standard urinary tests seem to be insufficiently specific and sensitive for the assessment of IgAVN development and activity because, according to some authors, even normal urinary findings do not exclude the possibility of nephritis activity and chronic IgAVN development. Kidney biopsy is an invasive procedure with possible complications and histological findings that indicate current kidney changes without nephritis occurrence or progression risk.
To provide additional clarifications about value of e-GST determination in patients with IgAV, we have expanded the Discussion section and added an evaluative list of conclusions.
Round 2
Reviewer 1 Report
Comments and Suggestions for Authors
This is a study of 97 patients with IgA vasculitis (IgAV) to evaluate the erythrocyte glutathione S-transferase (e-GST) activity. The authors demonstrated that e-GST activity levels were significantly elevated in the group of IgAV with nephritis (IgAVN) than those in groups of IgAV without nephritis (IgAwN) and controls. The presented study was well performed, and the revised manuscript is described in a reasonable manner.
Although the aim of this study was to evaluate that e-GST activity can be used as an additional sensitive marker of nephritis onset and activity in children with initially normal urinary findings, how many children with nephritis had normal urinary findings at the beginning, or cases of nephritis whose urinary findings temporarily normalized during follow-up? In addition, were there any differences in e-GST activity levels between IgAV with initially normal urinary findings and IgAV with abnormal urinary findings? Moreover, authors should additionally provide the excretion of urinary protein to the ROC curve analysis to clearly demonstrate the superiority of e-GST activity.
Author Response
Dear Sir / Madam
Thank you for your effort and time. Here are the responses to your questions/suggestions:
- At the beginning of the study, eight of all patients diagnosed with IgAVN had normal urinary findings (lines 98-104 and line 105)
- Two of all the patients diagnosed with IgAVN had temporarily normalized urine findings (abnormal findings at disease onset and 6 months after disease onset and normal findings after 3 months after disease onset) (line 106).
- No significant differences in e-GST activity were observed between IgAVN patients with initially normal urinary findings and IgAVN patients with initially abnormal urinary findings. (Table S4)
- We completely revised the ROC curve analysis.
Reviewer 2 Report
Comments and Suggestions for Authors
No further comments. The answers from authors are clear. The medical community in real life will decide whether the test is useful
Author Response
Dear Sir / Madam
Thank you for your effort and time.
Round 3
Reviewer 1 Report
Comments and Suggestions for Authors
This is a study of 97 patients with IgA vasculitis (IgAV) to evaluate the erythrocyte glutathione S-transferase (e-GST) activity. The authors demonstrated that e-GST activity levels were significantly elevated in the group of IgAV with nephritis (IgAVN) than those in groups of IgAV without nephritis (IgAwN) and controls. The presented study was well performed, and the revised manuscript is described in a reasonable manner. Authors had also responded to my all comments.